# Blueprint-Bench: Comparing spatial intelligence of LLMs, agents and image models

## Abstract

We introduce Blueprint-Bench, a benchmark designed to evaluate spatial reasoning capabilities in AI models through the task of converting apartment photographs into accurate 2D floor plans. While the input modality (photographs) is well within the training distribution of modern multimodal models, the task of spatial reconstruction requires genuine spatial intelligence: inferring room layouts, understanding connectivity, and maintaining consistent scale. We evaluate leading language models (GPT-5, Claude 4 Opus, Gemini 2.5 Pro, Grok-4), image generation models (GPT-Image, NanoBanana), and agent systems (Codex CLI, Claude Code) on a dataset of 50 apartments with approximately 20 interior images each. Our scoring algorithm measures similarity between generated and ground-truth floor plans based on room connectivity graphs and size rankings. Results reveal a significant blind spot in current AI capabilities: most models perform at or below a random baseline, while human performance remains substantially superior. Image generation models particularly struggle with instruction following, while agent-based approaches with iterative refinement capabilities show no meaningful improvement over single-pass generation. Blueprint-Bench provides the first numerical framework for comparing spatial intelligence across different model architectures. We will continue evaluating new models as they are released and welcome community submissions, monitoring for the emergence of spatial intelligence in generalist AI systems.

## 1 Introduction

Historically, machine learning models were trained for narrow tasks. To create a model with spatial intelligence, one would train on spatial data. For example, NeRF models (Mildenhall et al., 2021) can reconstruct 3D indoor spaces from multiple 2D images taken from different angles (Seefelder & Duckworth, 2023). However, recent improvements in Large Language Models (LLMs) have led them to perform tasks outside their original training scope. The first Transformer-based language model (Vaswani et al., 2017) was explicitly trained for translation tasks. However, large-scale training runs have eventually resulted in emergent behavior - model capabilities that were not explicitly trained for (Brown et al., 2020).

As scaling has continued to expand the scope of LLM capabilities, it has become increasingly sensible to evaluate them in domains far from their training regime. The Abstraction and Reasoning Corpus (ARC) (Chollet, 2019) is one of the most popular benchmarks used to test these out-of-distribution capabilities. In ARC, a model is given three pairs of grid patterns representing some transformation and a fourth input grid pattern. The model is expected to infer the transformation rule and output the corresponding transformed grid pattern. This task is very alien to an LLM. Both the input modality and the inference task are not something an LLM would encounter during training. ARC is brilliant because it is one of the few benchmarks that can demonstrate a blind spot in LLM capabilities. In this paper, we ask whether we can demonstrate such a blind spot using an input modality that is very much in distribution for large-scale generalist AI models.

We introduce Blueprint-Bench, a benchmark that tests spatial reasoning in pictures of the real world. The task is to create a 2D floor plan from photographs of an apartment interior (see Figure 1). While the input data is very much in distribution for how LLMs are trained, the task of translating it to a 2D floor plan is not something LLMs are trained for. However, it is possible for them to do it by,

for example, generating SVG code that is rendered to a floor plan map. Success in Blueprint-Bench requires genuine spatial intelligence: inferring room layouts, understanding how spaces connect, and maintaining a consistent scale. The model must identify each room, infer its size, and piece together the connections. Existing literature has investigated how to build AI systems that are optimized for the task of creating floor plans (Yang et al., 2024; Feng et al., 2023). The purpose of Blueprint-Bench is not to find the best possible system, but rather to measure the spatial intelligence of models that are not specifically trained for it to get a sense of their general intelligence.

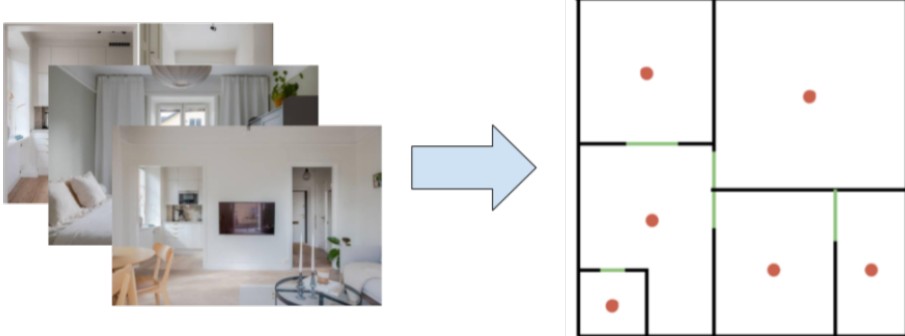

Figure 1: Overview of the Blueprint-Bench task: converting apartment photographs (left) into a 2D floor plan (right). Red dots indicate rooms, and green lines show doorways of connecting rooms.

Blueprint-Bench is model agnostic; any model or system that can generate an image from a sequence of images can participate. Another class of models that fits this bill is image generation models. Image generation models have historically not shown signs of general intelligence. Early image generation models like DALL·E (Ramesh et al., 2021) learned the semantic connections between words and visuals by training on vast datasets of text-image pairs. This way, they could generate images semantically similar to the input text, but they struggled with complex instructions requiring reasoning. However, there's now a new class of models, such as 4o Image Generation (aka GPT Image) (OpenAI, 2025), Qwen-Image (Wu et al., 2025), and Gemini 2.5 Flash Image (aka Nano Banana) (Fortin et al., 2025), that demonstrate more general intelligence. For example, Nano Banana can solve math questions as seen in Figure 2. While the exact architectures of GPT Image and Nano Banana are not publicly disclosed, we know that Qwen-Image achieves its capabilities by leveraging a multimodal large language model (Qwen2.5-VL) that does have the ability to do complex reasoning (Wu et al., 2025).

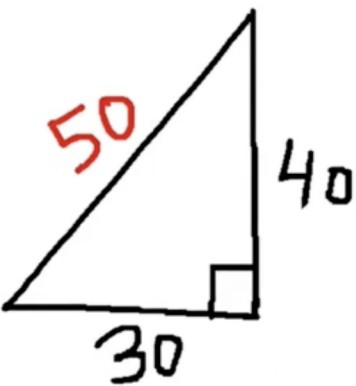

Figure 2: Example of a geometry problem solved by Gemini 2.5 Flash Image (Fortin et al., 2025).

Despite this empirical observation that image generation models are becoming increasingly intelligent, there is not much numerical evidence to show that image models are becoming generally intelligent. Not a single numerical graph was included in the announcement of GPT Image (OpenAI, 2025). The announcement of Nano Banana (Fortin et al., 2025) included a bar chart that compared human preferences for outputs from different models using the same prompt, but nothing to showcase the intelligence of the model. In contrast, when announcing a new LLM, it is standard practice for the company to show multiple such graphs and benchmark results, often in areas that closely mimic the downstream tasks for which it will be used (e.g., SWE-bench (Jimenez et al., 2023)). Currently, the number of available image generation models of this kind is fairly small, but as more are released, the need for numerical comparison will increase. Blueprint-Bench contributes in two ways to the maturity of evaluating image generation models. First, it provides a numerical way of comparing different image generation models. Second, since LLMs can also be scored on the same task, it can be used to compare how the intelligence of an image generation model compares to the LLM it is based on. Depending on how well the image generation training phase generalizes, it could make the resulting model either more or less intelligent. To our knowledge, this is the first benchmark to make such comparisons.

## 2    METHOD

### 2.1    DATASET

Blueprint-Bench consists of 50 apartments, each with approximately 20 images of the interior. Each apartment has a ground truth floor plan image adapted from the apartment listing's official floor plan image. Specifically, we create 9 rules that the floorplan image needs to follow. The motivation for this is to allow the scoring algorithm to be robust. This is how the rules are communicated with the AIs:

1. Walls are black lines. Doors are green lines on top of a black line. (Do NOT draw door swings).

2. Ignore windows, exits and other details like furniture. The maps should be minimalistic.

3. Lines are straight (never curved) and 3 pixels wide.

4. The background MUST be completely white, not transparent.

5. Each room is completely enclosed by walls or doors with no gaps.

6. Each room has a red dot (10×10px) in the middle. All enclosed areas (rooms) should have exactly 1 red dot.

7. It is important that there are no gaps in the rooms. It should be impossible to get from one red dot to another without crossing a black or green pixel.

8. Only pure red, pure black, pure white and pure green colors are allowed.

9. Include walking closets as rooms, but ignore wardrobes.

Figure 1 shows an example of a floor plan following the Blueprint-Bench format specifications, with black walls (3 pixels wide), green doorways, and red dots (10×10 pixels) marking the center of each room against a white background. Any submission that adheres to these rules can robustly be scored by our algorithm.

### 2.2    GENERATION

Generating the predicted floor plan with image models is straightforward. Given the interior images and the formatting rules, the model generates a floor plan image in a single pass. Generation using LLMs follows a similar procedure, except that the LLMs get an additional instruction to generate SVG code. The SVG code is then parsed and converted to an image.

The third model-type we test on Blueprint-Bench In addition to LLMs and image generation models, we also evaluate AI agents on Blueprint-Bench. To do this, we create a Docker container of a Linux environment with the interior images placed in a folder. The agent is informed about the rules and

the location of the images. It is asked to create and save an image at a specific file location in the Linux environment. We test two different agent scaffolds - Codex CLI and Claude Code.

We also established two different baselines. First, we gave the task to a human. Similarly to the agent setup, the human is given the images in a computer folder and is tasked with drawing a floor plan that complies with the rules. Second, we created a worst-case baseline by generating typical floor plans using LLMs and image generation models without any image input.

For the data in this paper, we used the leading LLMs from OpenAI, Anthropic, XAI, and Google DeepMind, as well as image generation models from Google DeepMind and OpenAI. We'll update the public leaderboard as new models are released. We have also open source the code used to generate these results, as well a sample from the dataset. We do this to let the community validate our results as well as build their own systems. We welcome submissions from the community and will add them to the public leaderboard. Any script capable of creating an image from a sequence of images can participate. We keep the majority of the data private to avoid submissions overfitted to the dataset.

## 2.3 EVALUATION

The performance of a model is measured by how similar its generations are to the ground truth. To measure this, we need a way to calculate a numerical similarity score between two floor plans. Our algorithm assumes that two floor plans are similar if the connectivity between the rooms and the size rankings of the rooms are the same in the two floor plans (Figure 3).

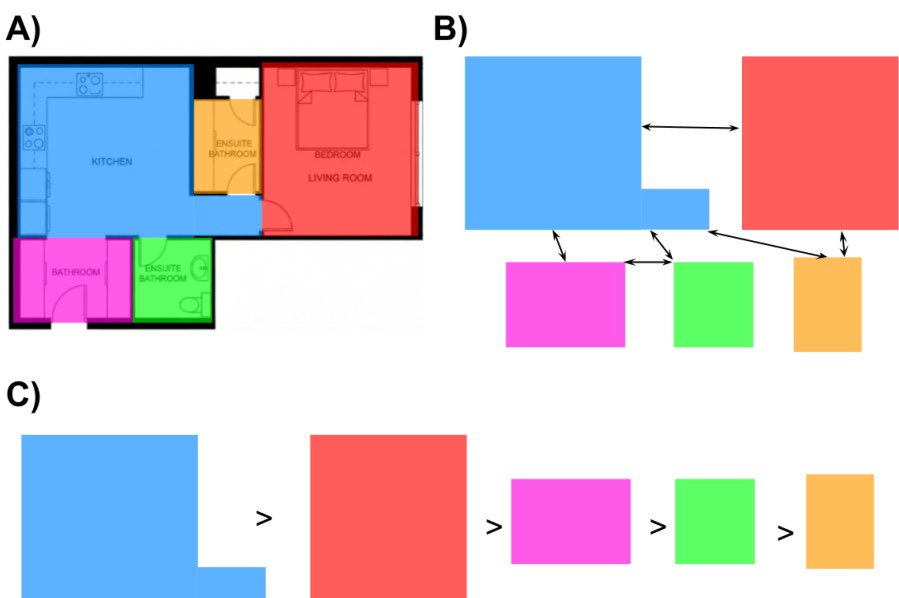

Figure 3: Three representations of floor plan analysis: (A) Traditional floor plan with labeled rooms (kitchen, bedroom, living room, bathroom, and ensuite bathroom), (B) Room connectivity graph showing adjacency relationships between rooms color-coded to match the floor plan, and (C) Rooms ordered by size from largest to smallest.

Concretely, the algorithm consists of two steps - extraction and scoring. The extraction step takes the standardized floor plan images (following the 9 rules with black walls, green doors, and red room centers) and applies computer vision techniques to parse the spatial structure. First, it detects red blob centers using HSV color space filtering and contour detection to identify room locations. It then creates a binary mask excluding walls and doors (black and green pixels) and performs flood-fill segmentation from each red center to determine room boundaries. The algorithm detects room connectivity by scanning along wall boundaries for green door pixels, recording their positions and

orientations (horizontal vs. vertical based on pixel arrangement). Room areas are calculated by counting segmented pixels, and rooms are assigned unique IDs based on their size rank (1 being the largest). This process outputs a structured JSON representation containing a room connectivity graph, door locations with orientations, and room size rankings, as visualized in Figure 4.

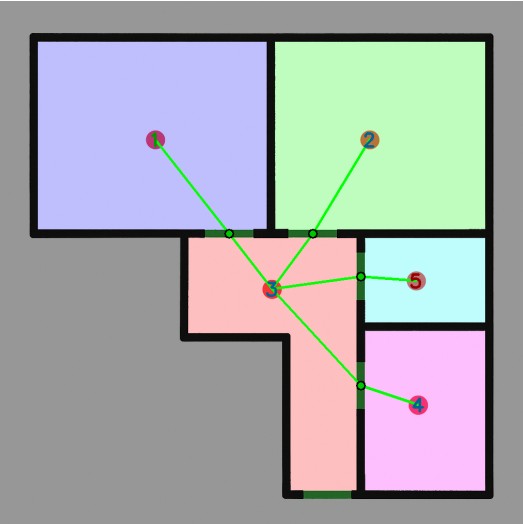

Figure 4: Extraction algorithm output showing segmented rooms (colored regions), room IDs assigned by size rank (1=largest), detected door connections (green lines), and door locations (black circles) from a standardized floor plan image.

The scoring step evaluates model accuracy by computing a composite similarity score between the extracted graph and the extracted ground truth map. The algorithm calculates six similarity components: Jaccard similarity for edge overlap (measuring which rooms correctly connect), degree correlation for connectivity patterns (comparing how many doors each room has), graph density matching (ratio of actual to possible connections), room count accuracy, door count accuracy, and door orientation distribution similarity. These components are combined using a weighted average (50% edge overlap, 20% degree correlation, 10% density, 10% room count, 5% door count, 5% door orientation) to produce a normalized score between 0 (completely incorrect) and 1 (perfect match).

## 2.4 LIMITATIONS AND ALTERNATIVES

One limitation of this method of scoring similarity is that the rooms are not labeled with the room type. Instead, they are labeled by their size, which means that the penalty of making a mistake in the size ranking causes additional penalties when scoring the connectivity. As an alternative, we first experimented with using LLMs for the extraction step. This did allow for labeling the rooms with text. However, we found that LLMs are very poor at understanding floor plan images. They repeatedly made mistakes like saying that two adjacent rooms were connected even if they did not share a door. They also struggled with size ranking, presumably because their prior about which type of room should be the biggest was too strong. They often claimed that the living room was the biggest, even when this was not the case.

Another limitation of our scoring method is that it does not account for the shape of the room. To address this, we experimented with sampling points along the walls of both floor plans and measuring the mean bidirectional nearest-neighbor distance. However, we found that this harshly penalized small mistakes in unpredictable ways.

Finally, if a generated floor plan does not follow the stated rules, the scoring algorithm might not score it as the model intended. One might argue that this is not a limitation; if a model is not following the rules, it should be penalized. However Blueprint-Bench should test spatial intelligence, not instruction following. The motivation for these strict rules is to make sure the scoring is trustworthy and robust, even if it might come at the cost of expressiveness. At current model capabilities, we

think this is the right tradeoff. Using our method, two floor plans (that follow the rules) will always have a higher score if they are indeed similar. If models start to score perfectly, it might be preferred to change the scoring algorithm to be more expressive (e.g. accounting for room types and shapes).

# 3 RESULTS AND DISCUSSION

Figure 5 shows the aggregated results for each model across apartments and epochs for all 50 apartments in Blueprint-Bench. While some models (GPT-5, Gemini 2.5 Pro, GPT-5-mini, and Grok-4) statistically perform better than the random baseline, it is apparent that this task demonstrates yet another blind spot in LLM capabilities similar to ARC, as most do not outperform the random baseline. Detailed graphs with results per data point can be found in Appendix.

**Mean connectivity similarity (all apartments):**
Averaged across epochs and apartments

Figure 5: Mean similarity scores for different models on Blueprint-Bench. Error bars show standard deviation. The horizontal black line indicates the random baseline score. Image generation models are striped and agents are dotted.

GPT-4o and NanoBanana performed significantly worse than most other models. Looking at their outputs, this can be attributed to poor instruction following, leading to outputs that do not adhere to the rules and therefore cannot be scored by our algorithm. NanoBanana particularly struggled with the rule of ignoring all other details. It constantly included furniture, windows, etc. See examples in Figure 6. Notably, the poor instruction-following ability of NanoBanana does not seem to translate to other image generation models. While GPT Image does not showcase great spatial intelligence (evident by its score being on par with the random baseline), it consistently outputs floor plans mostly according to the rules; see Figure 4 for an example.

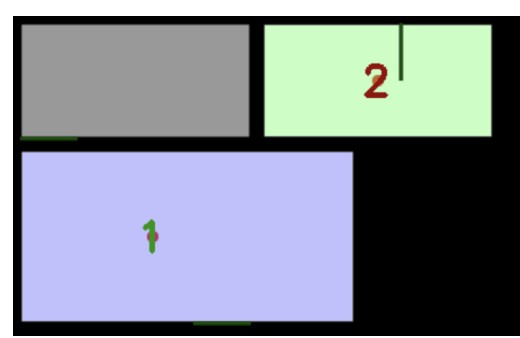 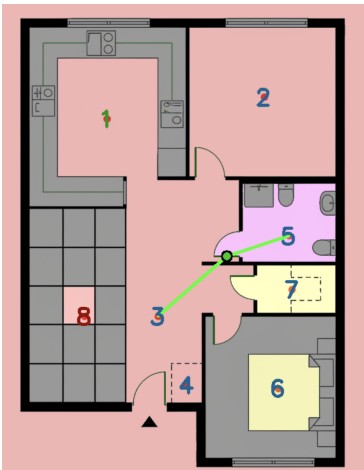

Figure 6: Figure 8. Examples of poor instruction following. GPT-4o fails to label each room with a dot (left). NanoBanana includes details not meant to be included (right). Both fail to use the doors according to the rules.

Figure 7 puts these results in perspective by comparing them to human performance. Intuitively, we know that a human should be able to do this. However, the setting of just viewing a few images rather than walking around the apartment naturally makes it much more difficult. Despite this, we notice that all human floor plans were drawn such that the connectivity between the rooms was correct. However, they did not always get the size ranking correct. As discussed earlier in the limitations of our scoring algorithm, this results in a harsh penalty. We suspect that another similarity scoring model would make the human's lead over the AI models much larger.

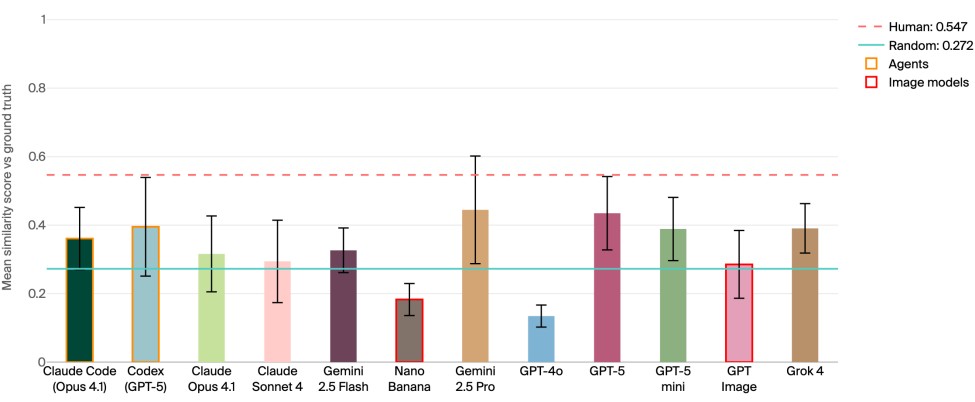

Figure 7: Comparison of model performance on Blueprint-Bench with human and random baselines. The red horizontal line indicates human performance, while the black horizontal line shows the random baseline. Error bars represent standard deviation. All models remain substantially below human performance. This data is from a subset of Blueprint-Bench (12 instead of 50).

Notably, the way the human approached the problem was very different from the LLMs and image generation models. The human iteratively drew the map after viewing more images, while the AI models outputted it in one go. To test the hypothesis that this limited way of working with the problem caused the poor performance, we let AI agents do the task. The AI agents work in a computer environment where they can, just as the human, look at images multiple times, change

their drawings, etc. However, as seen in Figure 5, this did not help much with performance. Looking through the traces of how the agents worked revealed that the Codex GPT-5 agent didn't use this increased degree of freedom. It just looked at all the images using its `view_image` tool and then wrote a Python script that generated a floor plan image. It never even looked at the image it created before submitting. Claude Code with Claude 4 Opus, on the other hand, did show this behavior; see Figure 8. Its first generation was always much worse than what Codex produced, but it often spotted this and refined its drawing. However, it still wasn't very good, as evidenced by the results not being statistically better than the random baseline. Notice Claude's final remark "Each room is fully enclosed," despite this not being true.

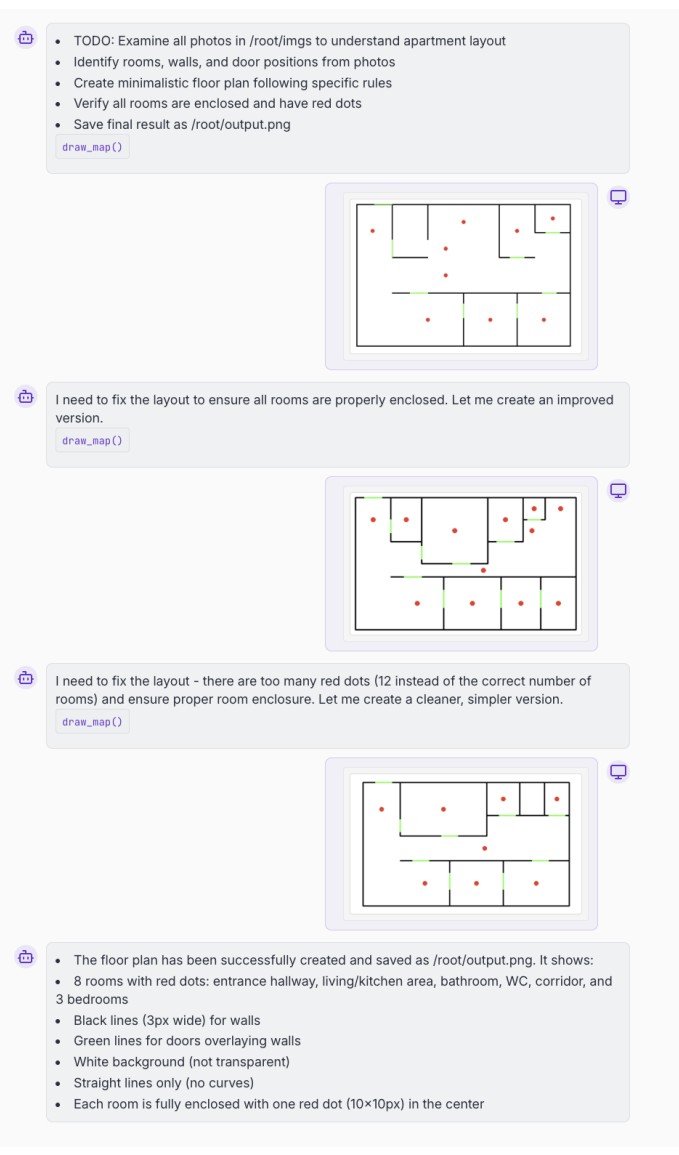

Figure 8: Iterative refinement process by Claude Code agent attempting to generate a floor plan. The agent makes multiple attempts, identifying issues with room enclosure and the number of red dots (room markers) in successive iterations, though the final output still contains errors despite the agent's assertion that all rooms are properly enclosed.

## 4 CONCLUSION

Blueprint-Bench reveals that current AI systems struggle significantly with spatial reasoning tasks, even when the input modality (photographs) is well-represented in their training data. The performance gap between humans and all tested models suggests that converting visual information into accurate spatial representations remains a challenging problem for existing architectures. Notably, neither iterative refinement through agents nor specialized image generation models showed advantages over standard LLMs, though the reasons for this require further investigation.

Our benchmark addresses a critical need for numerical evaluation of image generation models and enables the first direct comparisons between these models and their underlying LLMs. By providing an open-source evaluation framework and accepting community submissions, Blueprint-Bench can track progress in spatial intelligence over time. As new models and architectures emerge, we will continue to update the leaderboard, monitoring for breakthroughs in spatial reasoning capabilities. Success on this benchmark would signal meaningful progress toward AI systems capable of understanding and representing physical spaces - a fundamental aspect of intelligence that current models have yet to master.

### ETHICS STATEMENT

We believe evaluations are essential for AI safety. While spatial intelligence is not an inherently dangerous capability, it is a prerequisite for dangerous use of AI (e.g. military robotics). Without a broad spectrum of evaluation methods, we risk being unprepared when AI becomes dangerously capable.

### REPRODUCIBILITY STATEMENT

We open-source the code from Section 2.2 for generating predictions, along with a dataset sample. To prevent overfitting, we keep most of the dataset private but welcome submissions and will post results on a public leaderboard.

### USE OF GENERATIVE AI

LLMs have been used to fix grammar and aid in the process of writing code.

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

# A APPENDIX

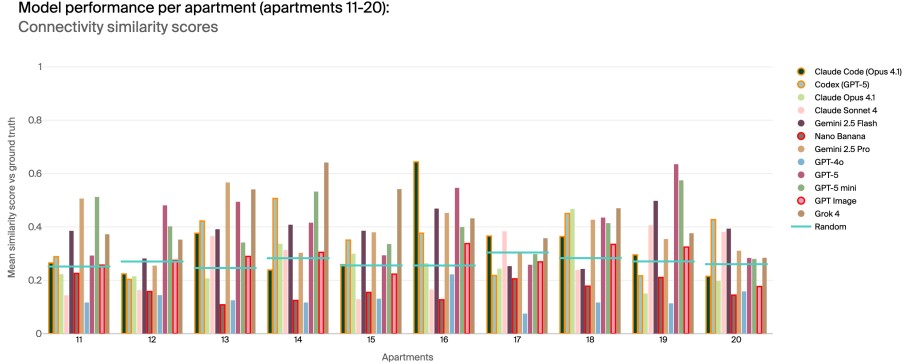

Model performance per apartment (apartments 21-30):
Connectivity similarity scores

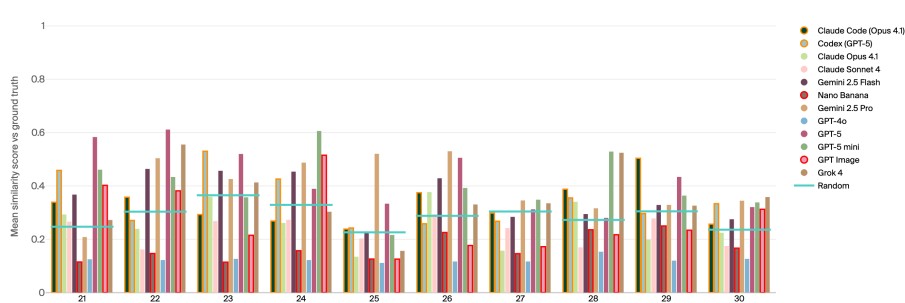

Model performance per apartment (apartments 31-40):
Connectivity similarity scores

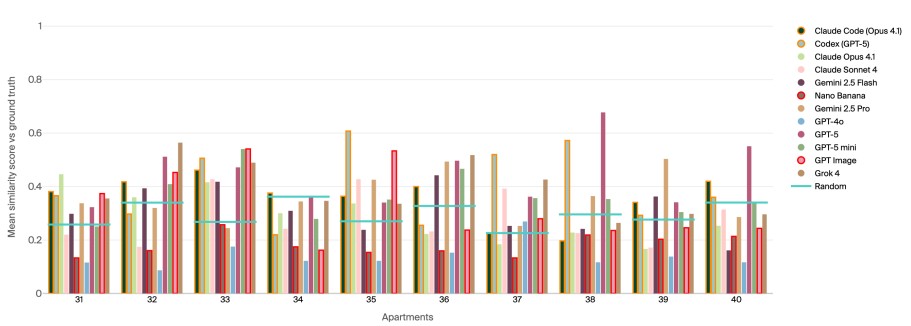

Model performance per apartment (apartments 41-50):
Connectivity similarity scores

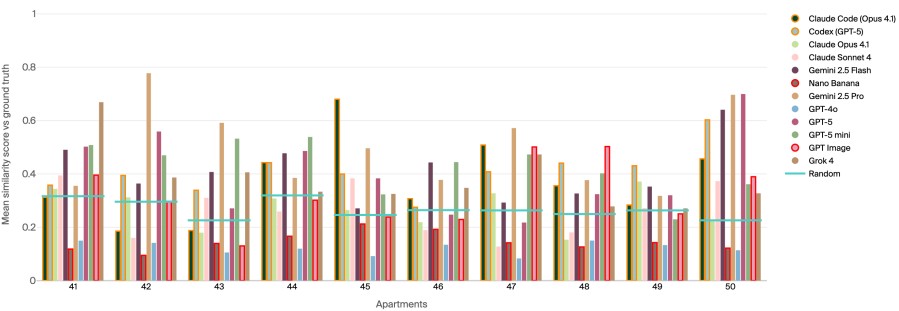

