# OpenReview forum: "Blueprint-Bench: Comparing spatial intelligence of LLMs, agents and image models"
_ICLR.cc/2026/Conference — Submitted to ICLR 2026_

### Official Review · Reviewer_TkLr · 2025-10-31

**Soundness:** 2
**Presentation:** 1
**Contribution:** 2
**Rating:** 0
**Confidence:** 4

**Summary:**

This paper introduces Blueprint-Bench, a new benchmark designed to evaluate the spatial reconstruction capabiility of general-purpose AI models. The task requires a model to generate an accurate 2D floor plan from a set of interior photos of an apartment. They frame this as a novel challenge where the input modality (photographs) is in-distribution, but the reasoning task (spatial reconstruction) is not.


Their contribution is: A new dataset of 50 apartments, each with multiple photos and a standardized ground-truth floor plan.

The authors evaluate a range of modern LLMs (GPT-5, Claude 4), image generation models (GPT-Image, NanoBanana), and agent-based systems (Codex, Claude Code). The results show a "significant blind spot" : most models perform at or below a random baseline, and all are substantially outperformed by a human baseline. The paper also finds that agent-based iterative refinement provides no meaningful improvement.

**Strengths:**

1. A new benchmark for Foundation models: The core idea is new (for foundation model community). The task of generating a floor plan from photos is intuitive, practical, and a clever proxy for evaluating spatial intelligence, especially spatial ability of transforming egocentric percpetions into allocentric representations.

**Weaknesses:**

1. Confounded Evaluation: The benchmark unfairly compares models by their output modality. It pits LLMs generating SVG code against image models generating raw pixels. This tests coding proficiency (for LLMs) and pixel-level instruction-following (for image models) as much as it tests spatial intelligence, making the cross-architecture comparison invalid.

2.Insufficient Data Scale: The dataset contains only 50 apartments. This sample size is too small to draw statistically significant or generalizable conclusions about the capabilities of large-scale models, especially when some analyses (like the human baseline) are run on an even smaller subset of 12 apartments.

3. Missing Numerical Results: The paper presents its primary findings exclusively through bar charts. It critically omits any tables of precise mean scores and standard deviations. This lack of exact numerical data hinders reproducibility and prevents a deeper analysis of the results.

4. Lack of analysis

5. Unprincipled "Random" Baseline: The worst-case baseline, which involves generating a floor plan "without any image input", is not a meaningful random baseline. It merely tests the model's internal prior for a "typical floor plan." A more principled baseline would involve randomizing the graph components being measured (e.g., generating graphs with random connectivity) to establish a scientifically meaningful floor for the scoring metric.

6. Vague Dataset Construction: The paper lacks critical details on dataset creation. While it mentions adapting "official floor plan images" , it fails to specify the data source (e.g., where the listings were from), the selection criteria for the 50 apartments, or the exact manual process used to "adapt" the ground-truth plans to the 9-rule format.

7. Missing Code and Evaluation Script: Despite the reproducibility statement , the code for the generation pipeline and, most importantly, the complex custom scoring algorithm  was not provided. This makes the benchmark unusable by the community and renders the results unverifiable.

**Questions:**

1. Dataset Construction: Could you please provide comprehensive details on the dataset construction? What was the source of the apartment listings (photos and original floor plans), and what was the manual adaptation process for creating the ground-truth images?

2. Output Modality: Given that the strict, rule-based image format primarily tests instruction-following , why not ask the models to output the spatial representation directly? For example, a JSON object describing the rooms, their size rankings, and their connectivity graph. This would provide a more direct measure of spatial intelligence, disentangled from generation-specific skills.

3. Agent Output Method: How exactly do the agent systems generate their final output? The paper states LLMs generate SVG , but the agent trace (Figure 8) implies the agent is writing and executing Python code to save a .png file. Is this correct? Does this different generation process (Python vs. direct SVG) affect the comparison?

---

### Official Review · Reviewer_8xX7 · 2025-11-01

**Soundness:** 2
**Presentation:** 2
**Contribution:** 2
**Rating:** 2
**Confidence:** 4

**Summary:**

This article introduces Blueprint-Bench, a new benchmark for evaluating the spatial intelligence of artificial intelligence models. Its core task is to generate accurate two-dimensional floor plans based on interior photos of apartments. Although the input is images (which are familiar to multimodal models), the output requires spatial reasoning capabilities (such as judging room layout, connectivity, and proportion consistency), which is highly challenging for most current AI models.

**Strengths:**

1. Introduced Blueprint-Bench, which includes 50 apartments, each with approximately 20 indoor photos. Each image has a corresponding real floor plan (obtained from property listings and standardized). The floor plans follow nine strict rules (such as walls being black lines, doors being green lines, and each room center having a red dot) to enable automatic scoring.
2. Evaluated three types of models: Large Language Models (LLM): such as GPT-5, Claude 4 Opus, Gemini 2.5 Pro, Grok-4. Image generation models: such as GPT-Image, NanoBanana. AI Agent systems: such as Codex CLI, Claude Code (capable of iterative operations in a Linux environment).
3. The scoring mechanism is novel: using graph structure similarity scoring: comparing the generated graph with the real graph in terms of room connectivity (graph edges) / room size ranking / number and direction of doors. The final score (0~1) is derived by integrating multiple indicators (such as Jaccard similarity, graph density, and door direction distribution).

**Weaknesses:**

1. The task of generating floorplans from real images is too trivial and does not address an essential spatial ability: Is it really measuring the model's understanding of spatial consistency? Or spatial cognitive ability? Or spatial memory?

2. Models for 3D scene layout generation have probably already done a good job. So why do this on models that have not undergone any specialized training?

3. If you want to test the spatial understanding of a unified model at the image generation level, generating floorplans is essentially generating an abstract top view. So why not directly generate a top view? Why not directly generate another perspective?

**Questions:**

1. The format is extremely non-standard and rudimentary, as if it were a student's final term report.
2. The analysis of the results is far from adequate, merely presenting the basic outcomes of different models and a meager number of examples (which seem to be screenshots).
3. Due to the various limitations of LLMs, the authors made many trade-offs, including using 'size' instead of 'room type' and not considering 'room shape'. I agree with this approach, but I believe connectivity is a concept that is highly correlated with room type. Furthermore, if only room size is considered, how is that different from just observing a single room from an ego-centric view and outputting its size?
4. How does the number and angle of input views (for each room) affect the results?

---

### Official Review · Reviewer_DYra · 2025-11-01

**Soundness:** 2
**Presentation:** 2
**Contribution:** 1
**Rating:** 2
**Confidence:** 4

**Summary:**

This paper introduces an innovative benchmark, Blueprint-Bench, designed to assess the spatial intelligence of current multimodal models by converting apartment photographs into accurate 2D floor plans. It evaluates three mainstream model categories: understanding models, generative models, and agent systems. Experiments conducted on 50 apartments (each with 20 interior images) demonstrate that current general-purpose models have yet to exhibit enough spatial intelligence.

**Strengths:**

- This work introduces a novel task that evaluates the spatial intelligence of multimodal models by converting apartment photographs into accurate 2D floor plans.
- The evaluation of agent systems and image generation models (particularly unified understanding-generation models like NanoBanana) is especially intriguing and relatively underexplored in prior studies.
- State-of-the-art models, such as GPT-5 and Grok-4, were tested, with understanding models tasked to generate SVG code.
- A well-designed metric was proposed to assess spatial reasoning, incorporating room connectivity graphs and size rankings.
- Experimental results reveal that current general-purpose models perform only marginally better than random baselines and fall significantly short of human-level performance.

**Weaknesses:**

Overall, the experiments are relatively limited, and the conclusions lack significant insights:

- The current experiments and analyses are insufficient to convincingly demonstrate that this benchmark reliably evaluates the spatial intelligence of models. For example, in the case of GPT-5, it is unclear why its performance is suboptimal—whether it struggles with understanding single image, establishing correspondence between multiple images, or other factors. Identifying specific shortcomings in existing models would better guide future developments in the field.
- The complementarity of this task with other benchmarks is not fully addressed. Undoubtedly, this is a novel task; however, it remains unclear whether it evaluates capabilities not addressed by existing benchmarks.
- Some experiments warrant deeper discussion. For instance, Figures 5 and 6 suggest that NanoBanana’s prompt-following ability is weak. However, does weak prompt-following necessarily imply poor spatial understanding? For example, if the task involved generating detailed BEV (bird’s-eye view) images, would NanoBanana perform better?
- As shown in Figure 7, considering the random baseline at 0.272 and the human baseline at 0.547, models such as GPT-5 and Gemini-2.5-Pro achieve scores in the 0.4+ range. What threshold should a model reach to be considered as demonstrating spatial intelligence?

Some claims also appear overly strong:

- The assertion that *“Blueprint-Bench provides the first numerical framework for comparing spatial intelligence across different model architectures”* is somewhat too strong, as several benchmarks aimed at quantifying multimodal models’ spatial intelligence have already been introduced this year [1, 2, 3].
- While Blueprint-Bench represents an important aspect of spatial intelligence and is a relatively comprehensive task, it does not encompass all aspects. As noted in [4], spatial intelligence includes diverse capabilities such as deformation and assembly, which are not reflected in this task.

A minor point: A similar idea is presented in Ross3D [5], where models are trained to enhance spatial understanding by generating BEV images from indoor videos. Although this differs from the benchmark, it may be worth some discussion.

[1] VSI-Bench: Thinking in Space How Multimodal Large Language Models See, Remember and Recall Spaces

[2] SITE: towards Spatial Intelligence Thorough Evaluation

[3] CoreCognition: Core Knowledge Deficits in Multi-Modal Language Models

[4] Holistic Evaluation of Multimodal LLMs on Spatial Intelligence

[5] Ross3D: Reconstructive Visual Instruction Tuning with 3D-Awareness

**Questions:**

I am particularly concerned about whether this benchmark can guide the development of spatial intelligence and its complementarity with other benchmarks, as detailed in the Weaknesses section.

---

### Meta-Review · Area_Chair_2sxc · 2025-12-22

**Summary:**

This paper presents blueprint-bench, a benchmark to evaluated the spatial intelligence of AI models. This is done through the task of converting apartment photographs into an accurate 2D floorpan. The authors evaluate various models, including both VLMs, image generation models, and agent systems. They find that models are consistently bellow human performance on this task.

**Reviewer Concerns:**

Reviewers were concerned about the efficacy of the benchmark, the validity of the claims, and the presentation of the work. I believe all of these concerns are still outstanding.

**Reviewer Scores:**

Given that the authors provided not rebuttal, I do not anticipate that the scores of any reviewer would have changed.

---

### Decision · Program_Chairs · 2026-01-26

Reject